# Polymer gels with tunable ionic Seebeck coefficient for ultra-sensitive printed thermopiles

Dan Zhao[1], Anna Martinelli[2], Andreas Willfahrt [1,3], Thomas Fischer[3], Diana Bernin[2], Zia Ullah Khan[1], Maryam Shahi[4], Joseph Brill [4], Magnus P. Jonsson [1], Simone Fabiano [1] & Xavier Crispin [1]

Measuring temperature and heat flux is important for regulating any physical, chemical, and biological processes. Traditional thermopiles can provide accurate and stable temperature reading but they are based on brittle inorganic materials with low Seebeck coefficient, and are difficult to manufacture over large areas. Recently, polymer electrolytes have been proposed for thermoelectric applications because of their giant ionic Seebeck coefficient, high flexibility and ease of manufacturing. However, the materials reported to date have positive Seebeck coefficients, hampering the design of ultra-sensitive ionic thermopiles. Here we report an "ambipolar" ionic polymer gel with giant negative ionic Seebeck coefficient. The latter can be tuned from negative to positive by adjusting the gel composition. We show that the ion-polymer matrix interaction is crucial to control the sign and magnitude of the ionic Seebeck coefficient. The ambipolar gel can be easily screen printed, enabling large-area device manufacturing at low cost.

[1] Laboratory of Organic Electronics, Department of Science and Technology, Linköping University, Norrköping SE-60174, Sweden. [2] Department of Chemistry and Chemical Engineering, Chalmers University of Technology, Gothenburg SE-41296, Sweden. [3] Innovative Applications of The Printing Technologies, Stuttgart Media University, Stuttgart 70569, Germany. [4] Department of Physics and Astronomy, University of Kentucky, Lexington KY40506-0055, USA. Correspondence and requests for materials should be addressed to S.F. (email: simone.fabiano@liu.se) or to X.C. (email: xavier.crispin@liu.se)

Thermoelectric materials enable direct conversion of heat to electrical signals and can be used for heat flux and temperature sensing[1]. These technologies are based on the Seebeck effect, i.e., the creation of a voltage across a material subject to a temperature gradient. The voltage originates from the diffusion of mobile charge carriers transported by the heat flux. The magnitude of this phenomenon is different in different materials, which can then be classified according to their Seebeck coefficient ($\alpha$). Different from conductivity ($\sigma$), which is proportional to the charge carrier concentration, the Seebeck coefficient typically decreases with increasing the concentration of charge carriers. Therefore, high Seebeck coefficients of $1 \, mV \, K^{-1}$ or higher are likely to be found in insulator materials having large energy gaps[2]. However, the low electrical conductivity of insulators (lower than $10^{-12} \, \Omega \, cm^{-1}$) makes it very challenging to perform reliable thermovoltage measurements. There is yet another special class of electronic insulators emerging for thermoelectric applications: electrolytes. Instead of electronic charge carriers, these materials have ionic charge carriers, which thermodiffuse under a temperature gradient through the Soret effect[3]. Recently, several groups reported extraordinary high values of ionic Seebeck coefficient in electrolytes, reaching $+10 \, mV \, K^{-1}$, and their ionic conductivity is large enough to ensure easy thermovoltage measurements[4–6].

One of the main differences between ionic and electronic thermoelectric materials is that ions cannot pass into an external circuit when reaching a metal electrode. Hence, ionic thermoelectric devices will not generate a constant electrical power when operated at a constant temperature difference[7]. However, the thermodiffusion of ions produces a large constant voltage that can be used for heat flux sensing or temperature measurements. Recent studies also indicate that the ions accumulated at the electrode/electrolyte interface can be exploited for energy conversion applications[8,9]. Indeed, the use of high capacitance electrode materials, such as carbon nanotubes and conductive polymers, dramatically enhances the amount of charges that can be accumulated at the electrode/electrolyte interface, thus allowing for charging supercapacitors[4,5,7] and batteries[10]. Polymer electrolytes with giant Seebeck coefficient have sparked interests also in new research directions, such as in thermoelectronic circuits employing heat as input signals, and ultra-sensitive temperature sensors competing with pyroelectric detectors[11]. Polymer-based electrolytes are attractive because they are solid (or gels) rather than liquids, which is advantageous when manufactured and used in iontronic devices. One can foresee possible application of this giant ionic Seebeck coefficient in combination with other polymer electrolyte-based devices, such as electrochromic displays[12], ion pumps[13], ionic bipolar diodes[14], ion bipolar junction transistors[15], and electrochromic bipolar membrane diodes[16].

The ionic Seebeck coefficient ($\alpha_i = V_{thermal}/\Delta T$) of an electrolyte is measured from the open circuit thermovoltage ($V_{thermal}$) induced over the material by a given temperature difference ($\Delta T$). The $V_{thermal}$ is proportional to the difference in concentration profile between anions and cations. In turn, the concentration gradients for anions and cations depend on the self-diffusion coefficients, the effective ionic concentrations (which only include the dissociated ions) and the Soret coefficients of the ions. The Soret coefficient is a relatively complex parameter determined by the temperature dependence of the structural entropy, which is related to interactions between ions and solvent along the thermal field[17]. Ions that increase the local order of the surrounding solvent molecules are named structure makers (kosmotropic effect), while those that reduce local order are named structure breakers (chaotropic effect)[18]. Modern theoretical modeling tools based on non-equilibrium molecular dynamics succeeded to reliably calculate the Soret and Seebeck coefficients in simple atomic salts such as LiCl[19]. In contrast, for polymer electrolytes, there is so far no satisfactory theory that can accurately describe or predict those coefficients. The intuitive strategy of choosing electrolytes with a large difference in diffusion coefficients between their anions and cations has led to extraordinary large Seebeck values, which cannot yet be fully explained. Examples include the solution of tetrabutylammonium nitrate in alcohols ($+7 \, mV \, K^{-1}$)[6], ionic functionalized liquid polyethylene glycol ($+11 \, mV \, K^{-1}$)[4], hydrated polystyrene sulfonic acid ($+8 \, mV \, K^{-1}$)[5], polystyrene sulfonate sodium ($+4 \, mV \, K^{-1}$)[7], and its composite with cellulose ($+8.4 \, mV \, K^{-1}$)[20]. Notably, all those giant Seebeck coefficients are positive, meaning that cations thermodiffuse easier than anions (in analogy to solid-state semiconductors, we define these electrolytes as "p-type"). However, effective thermoelectric modules rely on both positive and negative thermoelectric legs (p- and n-legs). While negative Seebeck coefficients of about $-1$ to $-2 \, mV \, K^{-1}$ have been reported for electrolytes undergoing electrochemical reactions at the electrodes (i.e., thermogalvanic effect)[21,22], no "n-type" ionic thermoelectric materials based on pure Soret effect have been reported to date. Therefore, in order to enable a powerful technology based on the ionic Seebeck effect, it is crucial to develop "n-type" polymer gel electrolytes with negative giant Seebeck coefficients.

Here, we present an "ambipolar" polymer gel with negative Seebeck coefficient and demonstrate that both the sign and the magnitude of the Seebeck coefficient can be controlled by tuning the composition of the polymer matrix. Pulsed field gradient (PFG) nuclear magnetic resonance (NMR) spectroscopy, Raman and infrared spectroscopy are used to investigate the motion and interaction of ions. Finally, we exploit the complementarity of "n-type" (negative $\alpha_i$) and "p-type" (positive $\alpha_i$) polymer gels to fabricate an ionic thermoelectric module by printing technology and demonstrate its temperature sensing function as an ionic thermopile.

## Results

### Thermoelectric characterization of pure ionic polymer gel.
We chose the copolymer poly(vinylidene fluoride-co-hexa-fluoropropylene) (PVDF-HFP) as a solid, water-free polymer electrolyte matrix (Fig. 1a). PVDF-HFP is semicrystalline in its neat form and possesses two phases: one rich in PVDF crystallites, that provides the mechanical strength to the copolymer, and the other phase rich in HFP which is amorphous and can be swelled with various polar solvents acting as plasticizers[23]. As electrolyte, we used the ionic liquid (IL) 1-ethyl-3-methylimidazolium ([EMIM]) bis(trifluoro-methylsulfonyl)imide ([TFSI]). The ionic conducting polymer gel composed of [EMIM][TFSI] and the PVDF-HFP matrix is chemically and thermally stable up to ~300 °C[24].

Figure 1b presents the ionic conductivity and Seebeck coefficient of pure [EMIM][TFSI] and [EMIM][TFSI]/PVDF-HFP polymer gels with different IL content in the polymer matrix (details can be found in Methods). The pure IL has an ionic conductivity $\sigma_i = 9 \, mS \, cm^{-1}$ (open blue square, similar to previously reported values[25]) and an ionic Seebeck coefficient of $-0.85 \, mV \, K^{-1}$. The ionic conductivity of the polymer gels containing IL increases significantly with increasing IL content, from $\sigma_i = 0.1 \, mS \, cm^{-1}$ for a weight ratio $W_{IL}/W_{PVDF-HFP} = 1$ to $\sigma_i = 6 \, mS \, cm^{-1}$ for $W_{IL}/W_{PVDF-HFP} = 8$ (Fig. 1b and Supplementary Figure 1), mainly due to an increase in charge carrier concentration, as well as to the IL plasticizing effect that reduces the polymer glass transition temperature[26]. In contrast, the ionic Seebeck coefficient averages to a value of about $\alpha_i = -4 \, mV \, K^{-1}$ and is roughly independent of the IL content in the polymer gel (Fig. 1b and Supplementary Figure 2). Note that $\alpha_i$ for

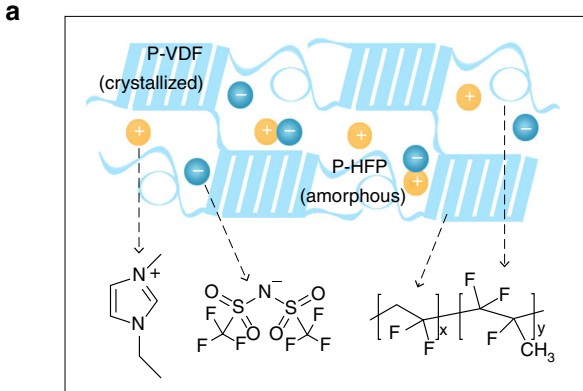

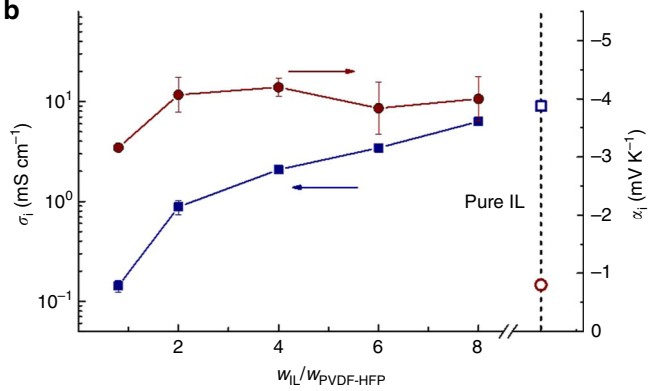

**Fig. 1** Composition and thermoelectric property of [EMIM][TFSI]/PVDF-HFP polymer gels. **a** Schematic illustration of the polymer gel composition. **b** Ionic conductivity (blue squares) and Seebeck coefficient (red dots) of the polymer gel for different weight ratio of [EMIM][TFSI] vs. polymer matrix PVDF-HFP ($W_{IL}/W_{PVDF-HFP}$). Bars represent mean ± s.d.

$W_{IL}/W_{PVDF} = 1$ is <20% lower compared to the other weight ratios. The time needed to reach a stable thermovoltage is about 200–300 s. Interestingly, the amount of ions needed to build the thermovoltage is much lower than the amount of ion contained in the gel for all the IL concentrations investigated here in this study (see Supplementary Note 1). For the pure IL, where both cations and anions thermodiffuse, the measured ionic Seebeck coefficient is smaller (open red circle in Fig. 1b, $-0.85\,\mathrm{mV\,K^{-1}}$). The enhanced negative sign of the Seebeck coefficient indicates that anions in the polymer gel thermodiffuse more easily than the cations. Since the Seebeck coefficient of electrolytes is closely related to the structural entropy induced by the interactions between ions and matrix[17,18], we hypothesize that the achieved negative value is due to the nature of PVDF-HFP. Due to the strong electron withdrawing character of the F-groups, the PVDF polymer backbone adopts a beta phase in the presence of ILs[27,28], which results in a high polarity chain. Together with a tendency to form narrow IL-rich nano-domains[23], the polymer matrix is likely to increase the local order (kosmotrope effect) and provide the anions with a higher thermodiffusion than cations. In order to evaluate the critical role of PVDF-HFP, we replaced it with the non-fluorinated block copolymer poly(styrene-block-methyl methacrylate) (PMMA-PS), featuring insoluble polystyrene PS domains and swollen PMMA domains when mixed with [EMIM][TFSI][29]. This gel shows a positive ionic Seebeck coefficient of $1.9\,\mathrm{mV\,K^{-1}}$ (Supplementary Figure 3). Nevertheless, the exact nature of ionic thermodiffusion in non-aqueous systems has been scarcely studied, and it is a challenging open question, which needs further theoretical and experimental investigations to be fully answered.

**Self-diffusion coefficient and conductivity measurements.** In order to tune the interaction between the IL and the fluorinated polymer matrix, we doped the polymer gel with low molecular weight ($M_w = 400$ Da) liquid polyethylene glycol (PEG). PEG has a lower $T_g$ and a higher segmental motion than PVDF, and it also plays the role of a Lewis-base that interacts preferentially with cations[30]. In all the following measurements, we have chosen polymer gels with a weight ratio $w_{IL}/w_{PVDF-HFP} = 4$, for which the polymer matrix is not saturated with the liquid component. Moreover, this ratio of IL/PVDF-HFP yields the highest negative Seebeck coefficient ($-4 \pm 0.2\,\mathrm{mV\,K^{-1}}$), decent ionic conductivity ($2 \pm 0.2\,\mathrm{mS\,cm^{-1}}$), and enough mechanical strength to be truly free-standing (Young's modulus of $\sim$1.3 MPa[25]).

We investigated the dependence of the self-diffusion coefficient of both ions and of the ionic conductivity on the amount of PEG (non-ionic) added to the polymer gel. The self-diffusion coefficients of cations ($D^+$) and anions ($D^-$) were determined by PFG-NMR spectroscopy (Fig. 2a, details can be found in Methods and Supplementary Figure 4). We found that both $D^+$ and $D^-$ initially increase for small amounts of added PEG (that is, when the molar concentration ratio $c_{PEG}/c_{IL}$ is increased from 0 to 0.052). Both $D^+$ and $D^-$ then decrease for higher molar concentrations of PEG ($0.052 < c_{PEG}/c_{IL} < 0.21$), reaching a plateau for $c_{PEG}/c_{IL} > 0.21$. This trend can be related to a change in viscosity in the local environment of the ions, which is however difficult to measure experimentally in a solid-like polymer gel.

From the measured self-diffusion values, it is possible to estimate the molar ionic conductivity $\Lambda_{NMR}$ (according to the Nernst–Einstein relation $\Lambda_{NMR} = N_A e^2 (D^+ + D^-)/kT$), which corresponds to the highest conductivity theoretically achievable assuming that all mobile ions are fully dissociated. In reality, however, a certain degree of ion pairing occurs which can be estimated by comparing $\Lambda_{NMR}$ with the molar conductivity measured by impedance $\Lambda_{imp}$ (see details in Supplementary Figure 5). Note that the $\Lambda_{imp}/\Lambda_{NMR}$ ratio is known as the ionicity of the system. As shown in Fig. 2b, $\Lambda_{imp}$ is lower than $\Lambda_{NMR}$ indicating some degree of ionic association, but displays an overall similar compositional dependence with a maximum at $c_{PEG}/c_{IL} = 0.052$. A further analysis of the $\Lambda_{imp}/\Lambda_{NMR}$ ratio (see Supplementary Figure 6) shows that the degree of ionic dissociation increases slightly in the composition range $0 < c_{PEG}/c_{IL} < 0.05$, a behavior that suggests a change in the local ion–ion and/or ion–polymer interactions. As the working mechanism, we postulate that PEG forms hydrogen bonds[31] with [EMIM] which disrupt the cation–anion pairs and promote ionic dissociation. Overall, as indicated in Fig. 2c, the initial increase in molar conductivity is attributed to an enhancement in ionic mobility as well as in the degree of dissociation (from i to ii), while the decrease in molar conductivity observed in the range $0.052 < c_{PEG}/c_{IL} < 0.21$ is mainly due to a lowered ionic mobilities (from ii to iii). For $c_{PEG}/c_{IL} > 0.21$ (iii), self-diffusion coefficients and ionic dissociation remain constant.

Raman spectroscopy reveals a red shift of the $C^{4,5}$-H stretching frequency at $\sim$3180 cm$^{-1}$ (Fig. 3a)[32], which is assigned to [EMIM] cations establishing H-bonds to the negatively charged oxygen atoms of PEG (more details can be found in Methods and Supplementary Figure 7)[33]. The strong Raman mode at $\sim$740.6 cm$^{-1}$ also shows a red shift with increasing PEG content (Fig. 3b), indicative of [TFSI] anions less strongly bound to their chemical environment[32]. These spectroscopic signatures are consistent with a specific PEG-[EMIM] interaction that interferes with the native [EMIM]-[TFSI] association (see Supplementary Figure 8). Figure 3b also shows an initial blue shift of the $\sim$740.6 cm$^{-1}$ mode that, although being of 0.3 cm$^{-1}$ only, turns into a red shift at $c_{PEG}/c_{IL} \approx 0.05$, a composition at which other properties show a trend change, e.g., the Seebeck coefficient

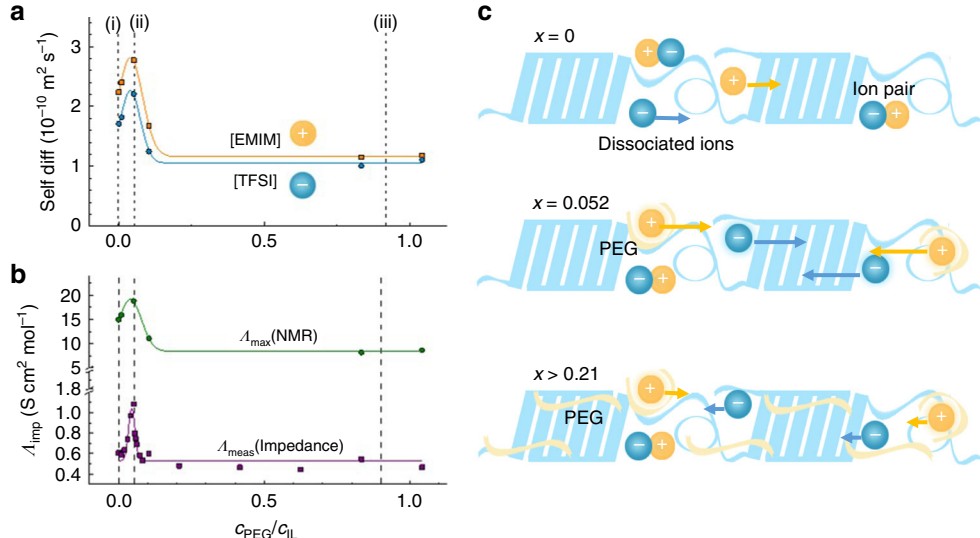

**Fig. 2** Self-diffusion coefficient of the ions and conductivity of the gels at different PEG content. **a** Self-diffusion coefficient of [EMIM] and [TFSI] vs. the molar concentration ratio of $c_{PEG}/c_{IL}$. **b** Molar conductivity calculated from impedance measurement ($\Lambda_{imp}$) and estimated from diffusion (PFG) NMR through the Nernst-Einstein relation ($\Lambda_{NMR}$), vs. molar concentration ratio of $c_{PEG}/c_{IL}$. **c** Sketch of the interaction between ions, PEG molecules, and polymer matrix

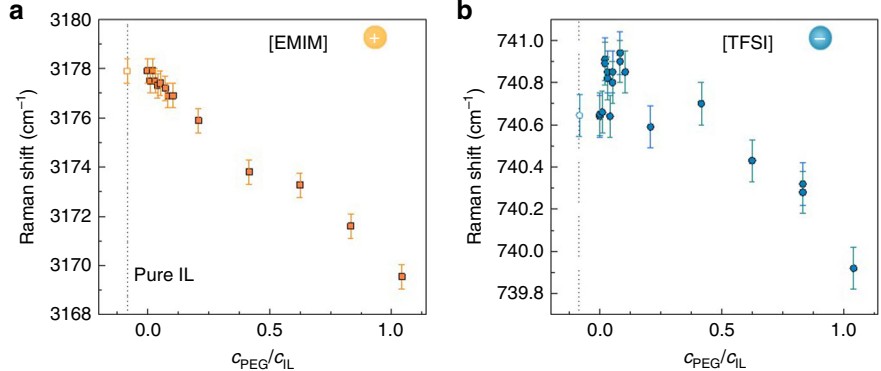

**Fig. 3** Characteristic Raman shifts of the gels with different PEG content. **a** $C^{4,5}$–H stretching mode of the [EMIM] cation and **b** expansion-contraction mode of the [TFSI] anion. Bars represent mean ± s.d.

changes its sign (see below). However, it is unclear whether the stronger association state of [TFSI] anions observed in the range $0 < c_{PEG}/c_{IL} < 0.05$ is due to interactions with PVDF-HFP only or if [TFSI]-PEG interactions also play a role. Alternatively, conformational changes or confinement effects limited to the 0–0.05 range may cause the observed behavior.

**Thermoelectric properties at different PEG content.** After providing insights on the interaction of ions with PEG molecules and the surrounding PVDF-HFP matrix, we focus on the effect of PEG on the Seebeck coefficient of the polymer gel. Figure 4a shows the ionic conductivity and Seebeck coefficient of the polymer gel as a function of PEG content. The Seebeck coefficient of the PEG-treated polymer gel first increases from about −4 mV K⁻¹ (at $c_{PEG}/c_{IL} = 0$) to almost 0 mV K⁻¹ (at $c_{PEG}/c_{IL} = 0.052$). A Seebeck coefficient close to zero indicates that both cations and anions contribute equally to the thermodiffusion. Interestingly, this occurs at the same composition that gives the maximum ionic conductivity, suggesting that both cations and anions contribute significantly to the charge transport at this IL-polymer gel composition. For higher PEG content ($c_{PEG}/c_{IL} > 0.052$), the Seebeck coefficient becomes positive and increases until it reaches

values as high as +13 mV K⁻¹, which is among the highest Seebeck coefficient reported to date.

Importantly, we observed a similar effect on the Seebeck coefficient when PEG is added to pre-formed IL-p gel films by pipette (instead of pre-mixing as shown above). As the PEG molecules permeate the polymer gel film, the ionic Seebeck coefficient switches sign, going from −4 mV K⁻¹ to +14 mV K⁻¹ (Fig. 4b and Supplementary Figure 9). As control experiment, we treated the non-fluorinated gel [EMIM][TFSI]/PMMA-PS with the same amount of PEG, and observed only a modest increase in the ionic Seebeck from +1.9 mV K⁻¹ to +3.5 mV K⁻¹ (Supplementary Figures 10). This supports our hypothesis that PEG favors the transport of cations. The compositional dependence on the Seebeck coefficient for this complex polymer gels can be interpreted by considering the interactions between PVDF-HFP and PEG with both types of ions. Figure 4c shows the interactional scheme for the polymer gels before and after PEG treatment. Without PEG, the PVDF-HFP matrix promotes thermodiffusion of anions over cations upon application of a temperature gradient, leading to a negative Seebeck coefficient. After PEG doping, the PVDF/anion interaction is hindered by the presence of PEG, while the interaction between cations and oxygen atoms contained in the PEG molecules promotes

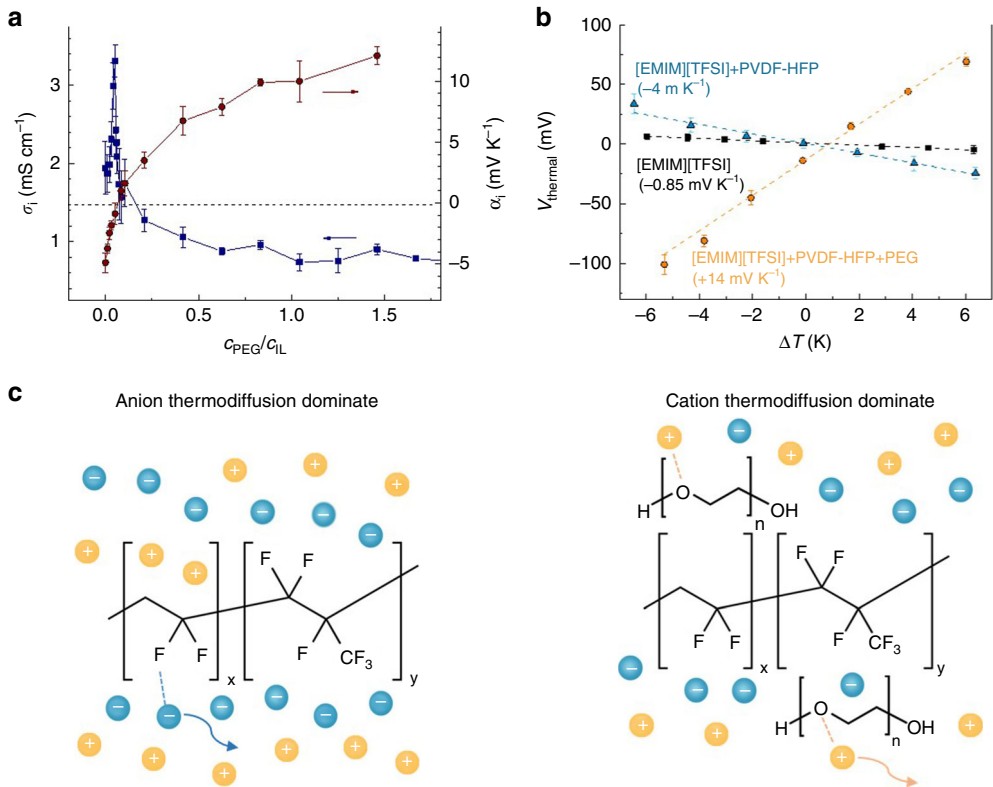

**Fig. 4** Thermoelectric properties of the gel at different PEG contents. **a** Conductivity (blue squares) and Seebeck coefficient (red dots, dashed line indicates Seebeck coefficient = 0 mV K$^{-1}$) of polymer gels as a function of the molar concentration ratio $c_{PEG}/c_{IL}$. **b** $V_{thermal}$ of pure IL, polymer gel, and polymer gel with PEG as a function of $\Delta T$. **c** Schematic illustration of the changing of Seebeck coefficient of the polymer gels. Bars in **a** and **b** represent mean ± s.d.

thermodiffusion of [EMIM] cations (and the material switches to a positive Seebeck coefficient).

**Ionic thermoelectric modules**. Next, we take advantage of the giant negative and positive Seebeck coefficients of our polymer gels to build an ionic thermoelectric module. Inspired by the strategy used in conventional electronic thermoelectric modules to increase the thermovoltage, we assembled electrically in series and thermally in parallel 18 n-legs of the polymer electrolyte [EMIM][TFSI]/PVDF-HFP and 18 p-legs of the PEG-treated [EMIM][TFSI]/PVDF-HFP. Figure 5a shows a schematic illustration of the device structure, with p- and n-legs drop-casted and connected by electrodes deposited on a glass substrate (a photograph of the device can be found in Supplementary Figure 11). The total Seebeck coefficient of the device was measured by heating the substrate to the different temperatures, and the corresponding temperature differences over the polymer gels ($\Delta T$) were measured across the sample while simultaneously recording the thermovoltage (see Fig. 5b). As shown in Supplementary Figure 11, $V_{thermal}$ increases linearly with $\Delta T$, yielding a slope as high as 0.333 V K$^{-1}$. The time needed to reach a stable thermovoltage is 150 to 250 s. This represents a record high value taking into account the very few number of legs. For comparison, typical commercial semiconductor thermopiles made of poly-silicon (p-Si) only provide 40 μV K$^{-1}$ per leg[34], which would require more than 8000 legs to match the thermovoltage of our device having only 36 legs. Notably, the $V_{thermal}$ generated by our ionic thermoelectric modules corresponds to 94% of the expected value calculated by summing the contribution of each leg (dashed line in Fig. 5b).

Benefiting from polymer electrolyte processing techniques developed for printed electronics[12], we then developed a fully printed polymer ionic thermoelectric module. More specifically,

we used screen printing to manufacture ultra-sensitive thermopiles based on the giant positive and negative Seebeck coefficients of our polymer gels. In addition to being low-cost, compared to traditional inorganic thermopiles, printed polymer ionic thermopiles also allow for flexible and large-area applications. The manufacturing is done in several steps, including printing the bottom electrodes, cavities defining the thermoelectric legs, and filling the cavities with the polymer gels (printing details can be found in Methods and a photograph of the printed devices on glass and flexible plastic substrates can be found in Supplementary Figure 12). The key strategy involves filling all cavities with the n-type polymer gel, then printing a small amount of PEG on every second leg, as illustrated in Fig. 5c. The PEG penetrates into the polymer gel and switches the sign of the Seebeck coefficient for those legs from negative (n-leg) to positive (p-leg). As shown in Fig. 5d, the final device produces a large thermovoltage, with a Seebeck coefficient of 182 mV K$^{-1}$ and a time needed to reach a stable thermovoltage of 150–250 s. With 40 thermoelectric legs, this corresponds to around 50% of the expected value (dash line in Fig. 5d). Considering that the manually fabricated device showed higher efficiency, we conclude that there is room for further optimization of the printing manufacturing process. For example, some of the legs might have become short-circuited during printing of the top electrodes, or the PEG may have diffused also into areas reserved for n-legs during the printing process. Importantly, this demonstration proves that polymer ionic thermoelectric modules can be mass produced using high-volume printing and extrusion technologies, which is a unique feature compared to conventional inorganic semiconductor thermopile and thermoelectric modules. Here we use metal electrodes to demonstrate the temperature/thermal detection applications of the ionic thermopile, while future devices may also utilize carbon electrodes, or conductive polymer electrodes, to

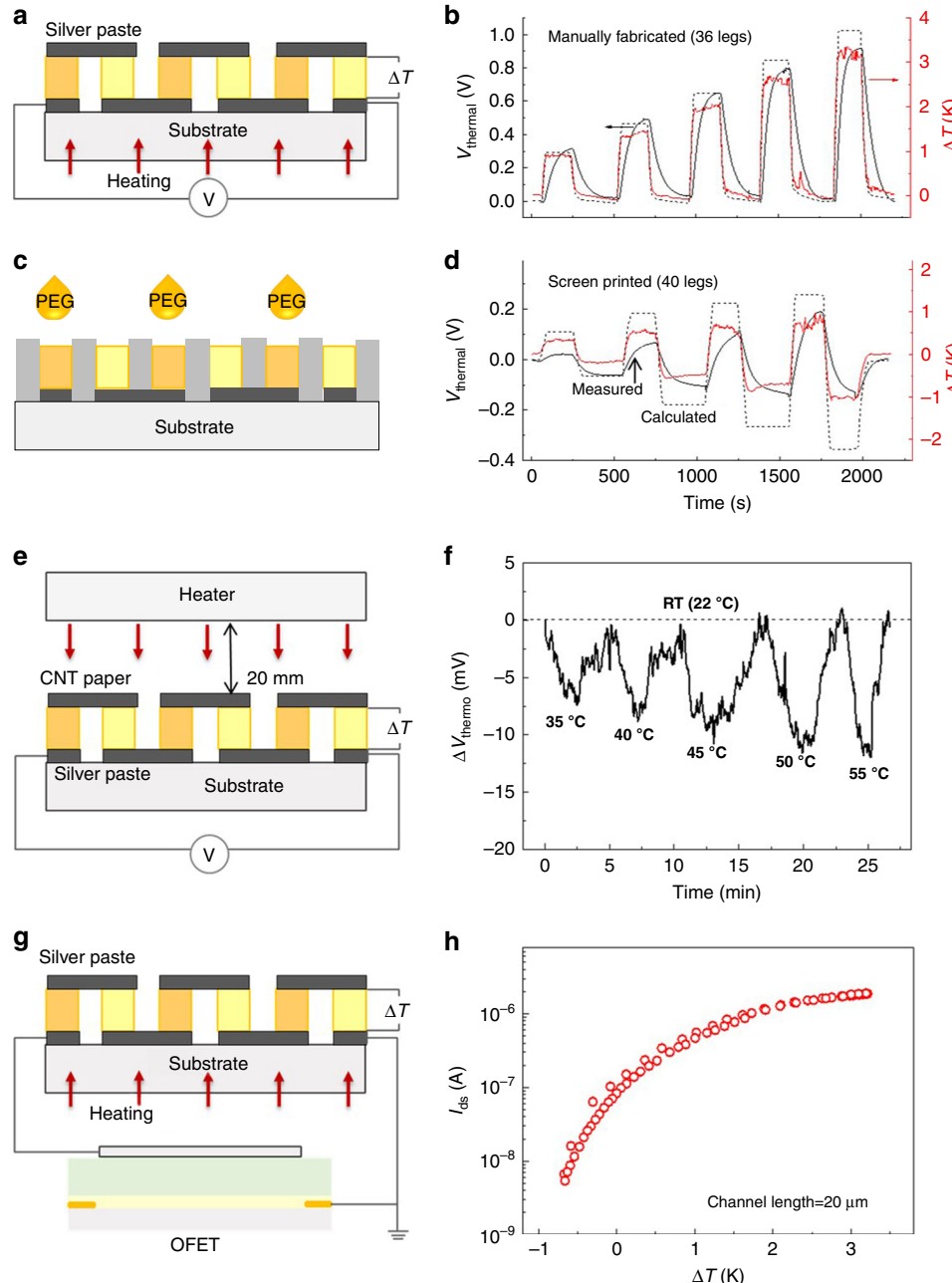

**Fig. 5 Ionic thermoelectric module. a** Schematic illustration of a manually made ionic thermopile and **b** $V_{thermal}$ changes with $\Delta T$ of a manually made ionic thermopile composed of 36 legs (thickness of $80 \pm 10$ μm, red solid line: $\Delta T$; black solid line: measured $V_{thermal}$; black dash line: calculated $V_{thermal}$). **c** Schematic illustration of the fabrication of a printed ionic thermopile and **d** $V_{thermal}$ changes with $\Delta T$ of a printed ionic thermopile composed of 40 legs. **e** Schematic illustration of the non-contact radiation heat detection and **f** device response to a hot object (Peltier heater) placed 20 mm away from the device and set at a temperature ranging from 35 to 55 °C. **g** Schematic illustration of an ionic thermoelectric gated transistor and **h** representative transfer curve (channel length = 20 μm) when $\Delta T$ is swept from −0.67 K to 3.2 K

store electrical energy via the concept of the ionic thermoelectric supercapacitor (ITESC)[4].

For practical application, a thermoelectric module should not require any external cooling element to establish a large $\Delta T$ when subjected to a heat source. Because of the comparatively low thermal conductivity of the polymer gel ($\kappa = 0.136$ W m$^{-1}$ K$^{-1}$, details can be found in Supplementary Note 2), a large $\Delta T$ across the polymer gel can be obtained even with the thickness of only a few tens of micrometers (Supplementary Figure 13). Combined with the large ionic Seebeck coefficient, the output voltage of the ionic thermopile can be greatly enhanced. In Fig. 5e, f, we report non-contact temperature detection by means of our 36-leg

polymer gel thermopile, using a Peltier heater as the heat source and placed 20 mm away from the thermopile top surface. The measured output voltage varied from 7 mV to 12 mV when the heater was set at a temperature ranging from 35 °C to 55 °C (Fig. 5f). For this measurement, as the top electrodes, we used a conducting paper composed of carbon nanotubes and nanofi-brillated cellulose (CNT-NFC, preparation details can be found in Methods) to increase the infrared absorption without excluding transferred heat.

The large Seebeck coefficient allows the ionic thermopile to provide a significant output voltage with only a few degrees of $\Delta T$, which can then effectively be used to gate a transistor[9]. Here we

show that a 36-leg ionic thermopile connected to the gate of an electrolyte gated field-effect transistor (Fig. 5g and Supplementary Figure 14) can modulate the source-drain current by more than two orders of magnitude by varying $\Delta T$ from $-0.6$ K to 3.2 K (Fig. 5h). Compared to the single-leg device reported previously[9], the detected temperature range is more than one order of magnitude smaller, translating into a $\Delta T$ sensitivity as low as 0.021 K. The high sensitivity of the ionic thermopile paves the way for heat- or infrared-gated electronic circuits with potential applications in photonics, thermography, and electronic-skins[35].

## Discussion

We demonstrate a non-aqueous polymer electrolyte gel that possess tunable giant ionic Seebeck coefficient. The sign and magnitude of the ionic Seebeck coefficient could be varied from $-4$ mV K$^{-1}$ to $+14$ mV K$^{-1}$ by tuning the composition of the "ambipolar" polymer electrolyte. We show that the addition of liquid neutral PEG to polyfluorinated co-polymers, swelled with ILs, can lead to a complete change in the dominating thermodiffused ions from anions to cations. Using the complementary of the ionic carriers, we build an ionic thermoelectric modules by connecting only 18 pairs of ionic thermocouples, showing a Seebeck coefficient of 0.333 V K$^{-1}$. In addition, we introduce the printed ionic thermopile as a new component for large area printed electronics by replicating the leg pattern through screen printing. The output voltage of traditional thermo-electric materials, typically used for temperature/heat sensing and energy conversion applications, is limited by their low Seebeck coefficient and high thermal conductivity. On the contrary, ionic polymer gels allow for improved $\alpha/\kappa$ ratio by 600 times. Ionic thermoelectric modules form a promising strategy for high-resolution temperature sensors beyond the limitations of traditional thermopiles and pyroelectric detectors, and their potential flexibility is a great advantage for human-organic electronic technology interfacing (i.e., e-skin), active flexible electronics and interactive buildings.

## Methods

**Materials**. Poly(vinylidene fluoride-co-hexafluoropropylene) (PVDF-HFP, $M_n = 130,000$), 1-ethyl-3-methylimidazolium bis(trifluoro-methylsulfonyl)imide ([EMIM] [TFSI]), PEG ($M_w = 400$ Da), Poly(styrene-block-methyl methacrylate) (PMMA-PS, $M_n = 32,000$ (PMMA: $M_n = 10,000$, PS: $M_n = 22,000$)), multi-walled carbon nanotube (m-CNTs, diameter: 5–9 nm, length: 5 μm) are purchased from Sigma-Aldrich and used as received. Nanofibrillated cellulose (NFC) was produced at Innventia AB, Sweden. Regioregular poly(3-hexylthiophene) (P3HT, Sigma-Aldrich) was dissolved in 1,2-dichlorobenzene (10 mg mL$^{-1}$) and filtered with a 0.2-μm polytetrafluoroethylene syringe filter. Poly(vinylphosphonic acid-co-acrylic acid) (P(VPA-AA), Rhodia) was dissolved in a mixture of 1-propanol and deionized water (40 mg mL$^{-1}$, solvent ratio of 4:1). The polyelectrolyte solution was then filtered with a 0.2 μm nylon syringe filter.

**Characterization**. The real part of the impedance and phase angle for all samples were measured by impedance spectrometer (Alpha high-resolution dielectric analyzer, Novocontrol Technologies GmbH, Hundsangen, Germany). An ac voltage of 5 mV was applied while sweeping the frequency from $1 \times 10^6$ Hz to $1 \times 10^2$ Hz. The measurements were carried out at room temperature. The ionic conductivity is calculated from $\sigma_i = \frac{L}{Z'A}$, where $Z'$ is taken at the frequency when the phase angle is closest to 0 (as shown in Supplementary Figure 1), $L$ is the distance between the two electrodes, and $A$ is the area of the electrodes.

The Seebeck coefficient of the IL-gels was measured at room temperature by alternating the temperature between the two electrodes (the setup is illustrated in Supplementary Figure 1). The voltage difference between the two electrodes was recorded by a nanovoltmeter ($\Delta V$) (Keithley Instruments, Inc., model 1282 A) and the temperature difference ($\Delta T$) between the two electrodes was evaluated using the thermocouples and measured using Keithley 2400 multimeter.

Diffusion (PFG) NMR experiments were performed on a 14.1 T Bruker Avance III HD ($^1$H Larmor frequency $-600$ MHz) spectrometer equipped with a Diff 30 probe head and GREAT 60 gradient amplifiers. Samples were prepared in NMR tubes and were kept at room temperature before starting any measurement. The experimental data was recorded at 22 °C using a stimulated echo pulse sequence with bipolar gradient pulses and spoilers. The gradient strength $g$ was ramped in 16 steps, the diffusion time was set to 50 ms and the gradient pulse duration $\delta$ to 1.5 ms. The apparent self-diffusion coefficient, $D$, was estimated by fitting the

Stejskal-Tanner equation $\ln(I/I_0) = -\gamma^2 g^2 \delta^2 (\Delta - \delta/3 - \tau/2)D$ to the echo signal decay $I$, where $I_0$ is the signal intensity at $g = 0$, $\gamma$ the gyromagnetic ratio, and $\tau$ the delay between the bipolar gradient pair. The calibration of the magnetic field gradient was performed at 22 °C using HDO in D$_2$O.

Raman spectra were recorded on an InVia Reflex Renishaw spectrometer using the 785 nm laser wavelength, a 1200 l/mm grating, a Peltier cooled CCD detector, and a x50 LWD Leica objective for the collection of back scattered light. The spectrometer was calibrated to the 1st order mode at 520 cm$^{-1}$ of a silicon wafer. Raman spectra were collected at room temperature. For quantitative analyses, Raman spectra in the frequency ranges 720–760 and 3000–3200 cm$^{-1}$ were peak-fitted using Gaussian functions and a linear background.

The FTIR (Fourier Transform Infrared Spectroscopy) spectrometer Equinox 55 from Bruker with the ATR (Attenuated Total Reflection) accessory A 225 having diamond crystal was used for studying the absorption of the polymer gels in the range of 400–4000 cm$^{-1}$.

**Preparation of the polymer gels**. The gels were prepared by mixing IL ([EMIM][TFSI]) and acetone solution of co-polymer (PVDF-HFP or PMMA-PS, $w_{polymer}/w_{acetone} = 1:7$) with different weight ratios. The mixture solution was stirred at least 30 min before the next step. For samples used in Seebeck coefficient and conductivity (impedance) measurements, the solution was drop-casted on substrates with pre-evaporated Au electrodes, dried in 60 °C oven for 15 min. For Raman and Infrared spectroscopy characterization and specific heat measurement, the films were prepared in the same way on glass substrates and then transferred after drying at 60 °C in an oven. For NMR characterization, the solutions of different IL-p gels were transferred into NMR tubes gradually in 2days, and kept in an oven at 60 °C.

**Screen printing of the device**. First the bottom silver electrodes covered with carbon-black are screen printed on the insulator substrate (glass or plastic) with designed pattern, and an UV cured resin was printed on top of the electrodes with cavity structure. Uniform IL-p gel (acetone solution, same recipe as described earlier) was then printed to fill the cavities. When dried, PEG modifier was printed to every second IL-p gel, thus to convert every second n-type leg to p-type leg. Finally, another layer of carbon paste was painted on top of the gel as the top electrode. The dimension of each printed leg is $2 \times 2$ mm$^2$.

**Preparation of the NFC-CNT electrodes**. A 0.3% (w/w) CNT stock solution was prepared, by blending water, CNT and 1% (w/w) Triton-X-100 in the ratio of 99.7:0.3:0.3 on the weight basis. The mixture was homogenized by using an IKA T-10 homogenizer, at 3000 rpm. The blend was thereafter sonicated in an ultrasonic bath (Ecoren BLC 8/3) at 50% power for 1 h. Further homogenization was achieved by an ultrasonic probe (Bandelin Sonopuls HD 2070) at 7 W, 20 kHz, for 5 min. Microfibrillated cellulose (1.9% (w/w) used as received), CNT (0.3% (w/w)), glycerol (the added amount was calculated as 10% (w/w) on the total amount of CNT and NFC), and water were mixed in 20 ml vials and homogenized with an Ultra Turrax homogenizer for 3 min. The vials were thereafter sonicated for 1 h. The content was diluted with water to a total volume of 30 ml, which was then homogenized for 1 additional minute with the Ultra Turrax.

**Preparation and characterization of the electrolyte gated transistors**. Inter-digitated source and drain electrodes (3-nm-thick Ti and 25-nm-thick Au) were prepared by photolithography on corning glass substrates pre-cleaned with deionized water, acetone, and isopropanol. P3HT was spin-coated from warm solution at 2000 rpm for 30 s, yielding a film thickness of about 30 nm. The films were then annealed at 120 °C for 30 min under nitrogen. P(VPA-AA) was then spin-coated at 2000 rpm for 60 s and dried on a hot plate under vacuum at 120 °C for 120 s, resulting in a 130-nm-thick film. A top gate titanium electrode with a thickness of 80 nm were deposited by thermal evaporation through a Ni shadow mask. The transistors were characterized using a semiconductor parameter analyzer (Keithley 4200-SCS).

## Data availability

The authors declare that the data supporting the findings of this study are available within the paper and its Supplementary Information Files.

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

## Acknowledgements

We acknowledge the Knut and Alice Wallenberg foundation (project "Tail of the sun"), the Swedish Foundation for Strategic Research (Synergy project), the Swedish research council (project "Next generation organic solar cells" and Grant No. 2016–03979, and Grant No. 2015-05070), Swedish Governmental Agency for Innovation Systems (Grant No. 2015–04859), the Swedish Energy Agency, the Advanced Functional Materials Center at Linköping University (Faculty Grant SFO-Mat-LiU No 2009 00971), the Swedish NMR Centre, the United States National Science Foundation Grant (DMR-1262261), VINNOVA (2015–04859), ÅForsk Foundation (18–351 and 18–313). We also acknowledge conceptualized.tech for helping with figures and images.

## Author contributions

D.Z., S.F., and X.C. conceived and designed the project. D.Z. fabricated and tested the ionic thermoelectric generators and performed the IR spectroscopy. Z.U.K. contribute to the Seebeck measurement setup and the measurement. D.B. and A.M. performed and analyzed NMR, Raman spectroscopy, and analyzed IR spectroscopy. A.W., T.F., and D.Z. performed the screen-printing of the ionic thermoelectric generators. M.S. and J.B. performed the specific heat measurement. M.P.J. simulated the heat distribution. D.Z., A.M., M.P.J., S.F., and X.C. wrote the paper. All authors contributed to discussion and manuscript preparation.

## Additional information

**Competing interests:** The authors declare no competing interests.

