## [Peer Review File · Nature Communications]

Reviewers' comments:

Reviewer #1 (Remarks to the Author):

The authors report a study on controlling ion transport in polymer blends to form materials that have positive and negative thermopowers. The resulting materials are used to form modules that provide a voltage under a thermal gradient and are used as a sort of sensor. The work is clearly reported, but it is difficult to tell if the results are a real fundamental advance.

There is a lot of work in the literature on studying ionic conduction in polymer blends with ionic liquids, particularly in the area of separators for batteries. It was not clear how the work in the manuscript compares to prior work, e.g. ionic conductivity, knowledge about ion-polymer interactions, etc. This makes it difficult to tell if there is a real fundamental advance.

There is some work in the literature, e.g.

Harvesting Waste Heat in Unipolar Ion Conducting Polymers
ACS Macro Lett., 2016, 5 (1), pp 94–98
DOI: 10.1021/acsmacrolett.5b00829

that discusses higher thermopowers in ionic systems, the role of humidity, etc.

Overall, the application in the manuscript, gating a transistor, has been shown by the authors previously. The relevance to other applications mentioned, e.g. thermopiles, etc. is not analyzed in any depth. It is therefore hard to know if the work here has any real advantage over existing thermal detection methods.

Reviewer #3 (Remarks to the Author):

This manuscript describes the development of ionic liquid-based gel for thermoelectric devices. The incorporation of an ionic liquid electrolyte into PVDF-HFP membranes results in a material, in which the Seebeck coefficient is tunable, a remarkable results that I am sure will interest workers in the area. The work has been carried out carefully and the manuscript has been prepared very well. I do think that it can be published in the journal. The area of research and the results of the work are sufficiently impactful to interest readers. My only concern is whether the manuscript is a little too long. I don't necessarily think there is extraneous material in it and it is concisely written. I guess I just defer to the editors here. If none of the other referees raises this issue, I am happy for it to go ahead without modification. In short I highly support publication.

Reviewer #4 (Remarks to the Author):

Review of Nature Communications Manuscript Number: NCOMMS-18-25221

The authors report the ability to control both the sign and magnitude of Seebeck coefficient of an ionic polymer gel by tuning the composition of the polymer matrix, along with the use of this material to fabricate both drop-cast and screen-printed ionic thermoelectric modules. In addition to the Seebeck coefficients of their materials and total Seebeck coefficient of the printed modules, the authors provide Raman spectroscopy of the materials to probe the bonding environment of the ions, and self-diffusion coefficients of the cations and anions as determined by pulse-field gradient nuclear magnetic resonance.

The tunability and characterization of these materials, along with the demonstration of its

impressive thermoelectric properties represents a significant advance for ionic thermopiles. This manuscript is both sufficiently novel, and relevant to a broad audience to justify publication in Nature Communications, after addressing the following comments:

1) Although the ability to tune the Seebeck coefficient of this polymer electrolyte brings yields interesting insight into the ionic diffusion and transport in the material, the authors' motivation for applications seems somewhat contrived. The author's state in the abstract that n-type "polymer electrolytes lag behind, hampering the design of ultra-sensitive ionic thermopiles." But the achievable positive Seebeck coefficient of their material is more than a factor of three larger than their negative coefficient, and so the thermopile sensitivity of a module based purely on their positive materials would have ~75% of the sensitivity of their p-type/n-type module. Later on page 5 of their manuscript, the authors state that "effective thermoelectric modules rely on both positive and negative thermoelectric legs." The authors should be specific here about what they mean with the word 'effective' and how much less 'effective' their module would be without an n-type leg, to demonstrate that development of n-type polymer gel electrolytes is "crucial".

2) On page 3, the authors state that the low electrical conductivity of insulators "makes it very challenging to perform thermo-voltage measurements with those materials. For this reason no thermoelectric modules have been developed based on insulators as active materials." The authors do not need to spend a large amount of effort justifying their materials, but should either provide references for these statements, or elaborate a bit further about why thermo-voltage measurements are challenging in insulators, and why (as they seem to suggest) it is the act of measuring the material properties, as opposed to the lack of electrical conductivity itself that has limited the development of insulators in thermoelectric modules.

3) On page 4, when the authors are discussing additional uses of this material for energy conversion applications, they state: "...when using suitable (high capacitance) electrode materials, such as carbon nanotube and conductive polymers, the accumulated charges can be dramatically enhanced...". The authors should be more precise in what they mean when they say 'enhanced' in this context. How is the 'enhancement' of a charge defined?

4) Although these time-scales can be inferred from other data presented by the authors, the authors should explicitly discuss the time constants for reaching the open-circuit voltage values for their measurements of Seebeck coefficient both for their materials, and modules.

5) The trend of Seebeck coefficient as a function of ionic liquid:matrix weight ratio shown in figure 2. b) appears to be non-monotonic. Additionally, there is a precipitous decrease in Seebeck coefficient as the weight ratio of ionic liquid to polymer matrix increases from 8:1 to the pure ionic liquid. The authors should elaborate on why this trend is non-monotonic, and why they believe the Seebeck coefficient of the ionic liquid is so much lower than with high weight ratios of the ionic liquid.

6) The figure caption for 2. b) repeats "(left axis)" when the second instance should read "(right axis)". Additionally, the left and right axes for this figure part could be contracted to magnify their data. for example, the Seebeck coefficient data could be displayed with an axis range of -2.5 mV/K to -5 mV/K, instead of 0 mV/K to -6 mV/K.

7) The authors sometimes use the term "thermovoltage" and other times use "thermo-voltage", and also sometimes use "thermodiffuse" and other times use "thermo-diffuse". They should take care to be consistent in their use of these terms throughout the manuscript.

Answers to the reviewer comments

Reviewer #1 (Remarks to the Author):

The authors report a study on controlling ion transport in polymer blends to form materials that have positive and negative thermopowers. The resulting materials are used to form modules that provide a voltage under a thermal gradient and are used as a sort of sensor. The work is clearly reported, but it is difficult to tell if the results are a real fundamental advance.

There is a lot of work in the literature on studying ionic conduction in polymer blends with ionic liquids, particularly in the area of separators for batteries. It was not clear how the work in the manuscript compares to prior work, e.g. ionic conductivity, knowledge about ion-polymer interactions, etc. This makes it difficult to tell if there is a real fundamental advance.

There is some work in the literature, e.g.

Harvesting Waste Heat in Unipolar Ion Conducting Polymers

ACS Macro Lett., 2016, 5 (1), pp 94–98

DOI: 10.1021/acsmacrolett.5b00829

that discusses higher thermopowers in ionic systems, the role of humidity, etc.

Overall, the application in the manuscript, gating a transistor, has been shown by the authors previously. The relevance to other applications mentioned, e.g. thermopiles, etc. is not analyzed in any depth. It is therefore hard to know if the work here has any real advantage over existing thermal detection methods.

We thank the reviewer for their thoughtful commentary and for giving us the possibility to strengthen our manuscript. The reviewer is correct in that polymer blends with ILs have been investigated for decades in the field of batteries, but very limited studies have focused on the development of ionic conductors for the emerging field of ionic thermoelectrics, which is the subject of the present study. We thank the reviewer for pinpointing the interesting work of Segalman et al. (ACS Macro Lett. 2016, 5, 94) that we had in good faith overlooked. We believe however that this previous study does not undermine the novelty of our work. Indeed, the phenomenon investigated by Segalman et al. in their work is a thermogalvanic effect, i.e. Ag^+ ions undergo an electron transfer at the Ag electrode ($\text{Ag} \rightarrow \text{Ag}^+ + \text{e}^-$). In our case, it is a true Soret effect of cations and anions. The ions chosen in our work do not undergo electron transfer at the electrode. The origin of the Seebeck coefficient in a thermogalvanic cell is related to the entropy variation upon the electron transfer; whereas the Seebeck coefficients in our study are due to the Soret coefficients of the charged species and are the highest reported to date for both p-type and n-type electrolytes.

The work by Segalman et al. is now included in the revised manuscript and commented at page 6.

We also wish to stress this is the first time that a drastic control of the transport properties of an IL, exemplified by the change in Seebeck coefficient and Λ , by addition of a secondary dopant is reported. We believe, the importance of these findings stretches beyond ionic thermoelectrics and will have implications in the field of battery as well. It will be interesting in fact for researchers working in the field of polymer electrolytes based on ILs to investigate how the Li^+ transference number may change in PEO/PVDF-HFP blends with composition, and also to understand if the properties of PEO and PVDF are additive (or competitive). So, we believe our findings have the potential to appeal to a broad audience.

Reviewer #3 (Remarks to the Author):

This manuscript describes the development of ionic liquid-based gel for thermoelectric devices. The incorporation of an ionic liquid electrolyte into PVDF-HFP membranes results in a material, in which the Seebeck coefficient is tunable, a remarkable results that I am sure will interest workers in the area. The work has been carried out carefully and the manuscript has been prepared very well. I do think that it can be published in the journal. The area of research and the results of the work are sufficiently impactful to interest readers. My only concern is whether the manuscript is a little too long. I don't necessarily think there is extraneous material in it and it is concisely written. I guess I just defer to the editors here. If none of the other referees raises this issue, I am happy for it to go ahead without modification. In short I highly support publication.

We thank the reviewer for considering our manuscript appropriate to Nature Communications. We checked the length of our manuscript and it is within the word limits for this journal.

Reviewer #4 (Remarks to the Author):

The authors report the ability to control both the sign and magnitude of Seebeck coefficient of an ionic polymer gel by tuning the composition of the polymer matrix, along with the use of this material to fabricate both drop-cast and screen-printed ionic thermoelectric modules. In addition to the Seebeck coefficients of their materials and total Seebeck coefficient of the printed modules, the authors provide Raman spectroscopy of the materials to probe the bonding environment of the ions, and self-diffusion coefficients of the cations and anions as determined by pulse-field gradient nuclear magnetic resonance.

The tunability and characterization of these materials, along with the demonstration of its impressive thermoelectric properties represents a significant advance for ionic thermopiles. This manuscript is both sufficiently novel, and relevant to a broad audience to justify publication in Nature Communications, after addressing the following comments:

We thank the reviewer for the very positive comments and for considering our manuscript appropriate to Nature Communications. In the following is a detailed answer to each point raised by the reviewer:

1) Although the ability to tune the Seebeck coefficient of this polymer electrolyte brings yields interesting insight into the ionic diffusion and transport in the material, the authors' motivation for applications seems somewhat contrived. The author's state in the abstract that n-type "polymer electrolytes lag behind, hampering the design of ultra-sensitive ionic thermopiles." But the achievable positive Seebeck coefficient of their material is more than a factor of three larger than their negative coefficient, and so the thermopile sensitivity of a module based purely on their positive materials would have ~75% of the sensitivity of their p-type/n-type module. Later on page 5 of their manuscript, the authors state that "effective thermoelectric modules rely on both positive and negative thermoelectric legs." The authors should be specific here about what they mean with the word 'effective' and how much less 'effective' their module would be without an n-type leg, to demonstrate that development of n-type polymer gel electrolytes is "crucial".

This is an excellent comment. From a manufacturing viewpoint, to screen print both p- and n-legs and then to connect them with alternating top and bottom electrodes is more effective than just using one type of legs. If only p-legs were used to build the thermoelectric module, the n-leg had to be replaced by a conductive material which would not contribute to the total output voltage. If a p-leg has a Seebeck coef of +5mV/K and a metal has a Seebeck coef -0.005mV/K, then one

thermocouple made of one p-leg and a metal leg gives 5.005mV/K. If a p-leg has a Seebeck coef of +5mV/K and a n-leg has a Seebeck coef of -5mV/K, then one thermocouple made of the p and n legs gives 10mV/K. In this way, the use of an n-type leg would make the manufacturing of ionic thermoelectric module more effective.

2) *On page 3, the authors state that the low electrical conductivity of insulators “makes it very challenging to perform thermo-voltage measurements with those materials. For this reason no thermoelectric modules have been developed based on insulators as active materials.” The authors do not need to spend a large amount of effort justifying their materials, but should either provide references for these statements, or elaborate a bit further about why thermo-voltage measurements are challenging in insulators, and why (as they seem to suggest) it is the act of measuring the material properties, as opposed to the lack of electrical conductivity itself that has limited the development of insulators in thermoelectric modules.*

We thank the reviewer for pinpointing this unclear statement. As correctly stated by the reviewer, the lack of electrical conductivity in insulators has hampered their development in thermoelectric modules for power generation applications. However, the focus of this manuscript is not on power generation but on thermal sensing. Insulators have a low electrical conductivity (typically lower than $10^{-12} \Omega \text{ cm}^{-1}$), which makes the thermovoltage measurement, performed using common voltmeters, unreliable. We have amended our statement in the revised manuscript (page 3).

3) *On page 4, when the authors are discussing additional uses of this material for energy conversion applications, they state: “...when using suitable (high capacitance) electrode materials, such as carbon nanotube and conductive polymers, the accumulated charges can be dramatically enhanced...”. The authors should be more precise in what they mean when they say ‘enhanced’ in this context. How is the ‘enhancement’ of a charge defined?*

Again, excellent comment. We were actually referring to the amount of charge (that is, charge density) that can be accumulated at the two different electrodes. As previously reported (Ref. 4, 5 of the revised manuscript), electrodes made of carbon nanotubes can induce accumulation of a much larger amount of charges as opposed to gold electrodes, for example. We now clarify our statement in the revised manuscript (page 4).

4) *Although these time-scales can be inferred from other data presented by the authors, the authors should explicitly discuss the time constants for reaching the open-circuit voltage values for their measurements of Seebeck coefficient both for their materials, and modules.*

We have added a discussion about the time constants in the supplementary information of the revised manuscript (page 7, 15).

5) *The trend of Seebeck coefficient as a function of ionic liquid: matrix weight ratio shown in figure 2. b) appears to be non-monotonic. Additionally, there is a precipitous decrease in Seebeck coefficient as the weight ratio of ionic liquid to polymer matrix increases from 8:1 to the pure ionic liquid. The authors should elaborate on why this trend is non-monotonic, and why they believe the Seebeck coefficient of the ionic liquid is so much lower than with high weight ratios of the ionic liquid.*

We thank the reviewer for giving us the possibility to clarify this point. The sudden decrease in ionic Seebeck coefficient as the W_{IL}/W_{PVDF} increases from 8:1 to pure IL is due to the preferential interaction of one of the ions with the surrounding PVDF-HPF matrix, as also stated in the original manuscript. In pure IL, both cations and anions can thermodiffuse so the measured ionic Seebeck coefficient is small. However, when the IL is combined with PVDF-HPF, due to the strong polar character of the PVDF chains, the polymer matrix is likely to favor the thermodiffusion of anions over cations. This results into large, negative ionic Seebeck coefficient values. It should also be noted that all Seebeck coefficient data for the IL/PVDF blend system are within error bars and averaged out at -4 mV/K, with the exception of that at $W_{IL}/W_{PVDF} = 1$ which is however less than 20 % lower compared to the others. We therefore argue that the ionic Seebeck coefficient is independent of the IL/PVDF blend composition over a wide range, whereas the ionic conductivity spans over two orders of magnitude.

6) *The figure caption for 2. b) repeats “(left axis)” when the second instance should read “(right axis)”. Additionally, the left and right axes for this figure part could be contracted to magnify their data. for example, the Seebeck coefficient data could be displayed with an axis range of -2.5 mV/K to -5 mV/K, instead of 0 mV/K to -6 mV/K.*

We thank the reviewer for spotting this typo, which we have corrected in the revised manuscript. As for the suggestion of displaying the Seebeck coefficient data on an axis range of -2.5 mV/K to -5 mV/K, this will unfortunately cut out the data relative to pure IL which has a Seebeck coefficient of about 0.8 mV/K (this was the reason why we originally chose to display the Seebeck coefficient data in the axis range 0 mV/K to -6 mV/K). We note however that all Seebeck coefficient data averaged out at -4 mV/K (see answer to comment 5 above), whereas the ionic conductivity spans orders of magnitude.

7) *The authors sometimes use the term “thermovoltage” and other times use “thermo-voltage”, and also sometimes use “thermodiffuse” and other times use “thermo-diffuse”. They should take care to be consistent in their use of these terms throughout the manuscript.*

We thank the reviewer for their sharp reading of the paper and have amended this mistake.

REVIEWERS' COMMENTS:

Reviewer #1 (Remarks to the Author):

The authors have addressed the question from the initial review. They should consider a balanced review of literature as suggested in the first review, for example there are a number of studies of similar effects that are not mentioned, e.g.

"Thermal Charging" Phenomenon in Electrical Double Layer Capacitors
DOI: 10.1021/acs.nanolett.5b01761

among others. It is still difficult to determine the advance of the paper without reference to other systems.

Reviewer #3 (Remarks to the Author):

I do think that the revisions made by the authors have increased the accessibility of the work. I still think that the manuscript is of sufficient interest and novelty for publication in the journal. The work seems very convincing to me and I am sure it will arouse a lot of interest among the readership.

Reviewer #4 (Remarks to the Author):

Thank you for your responses to the review. The manuscript is able to be published as is.

REVIEWERS' COMMENTS:

Reviewer #1 (Remarks to the Author):

The authors have addressed the question from the initial review. They should consider a balanced review of literature as suggested in the first review, for example there are a number of studies of similar effects that are not mentioned, e.g.

“Thermal Charging” Phenomenon in Electrical Double Layer Capacitors

DOI: 10.1021/acs.nanolett.5b01761

among others. It is still difficult to determine the advance of the paper without reference to other systems.

We thank the reviewer for their thoughtful comments. We have now included the suggested reference together with one more work (Qiao, Y., Punyamurtula, V. K., Han A. J., Thermally induced capacitive effect of a nanoporous monel, Appl. Phys. Lett. **91**, 153102 (2007)) at page 3 of the revised manuscript.

Reviewer #3 (Remarks to the Author):

I do think that the revisions made by the authors have increased the accessibility of the work. I still think that the manuscript is of sufficient interest and novelty for publication in the journal. The work seems very convincing to me and I am sure it will arouse a lot of interest among the readership.

We thank the reviewer for their thoughtful commentary of our manuscript.

Reviewer #4 (Remarks to the Author):

Thank you for your responses to the review. The manuscript is able to be published as is.

We thank the reviewer for their thoughtful commentary of our manuscript.